# The ecology and evolution of sub-exponential replicators

Bianka Kovács[1,2], György Barabás[1,3], Géza Meszéna[1,4], Eörs Szathmáry[1,2,5*], András Szilágyi[1,2]

**1** Institute of Evolution, HUN-REN Centre for Ecological Research, Budapest, Hungary, **2** Center for Conceptual Foundations of Science, Parmenides Foundation, Pöcking, Germany, **3** Division of Biology, Department of Physics, Chemistry and Biology (IFM), Linköping University, Linköping, Sweden, **4** Department of Biological Physics, Eötvös Loránd University, Budapest, Hungary, **5** Department of Plant Systematics, Ecology and Theoretical Biology, Eötvös Loránd University, Budapest, Hungary

* szathmary.eors@gmail.com

## Abstract

Systems of oligonucleotide chemical replicator molecules provide some of the finest, empirically realizable models of prebiotic evolution. Yet, a full understanding of their eco-evolutionary implications is hampered by conflicting assumptions, modeling strategies, and therefore predictions in the literature. Here we construct a model of these systems that accounts for the reversible association of templates and copies, ultimately leading to self-inhibition and sub-exponential growth. We show that, contrary to predictions from simplified model descriptions, there are well-defined limits on the attainable diversity of different replicator species. We also demonstrate that increasing the overall concentration of the system increases diversity, but counterintuitively, an analogous increase in the available resource concentration has the opposite effect. Most notably, if an exponentially-growing replicator is also present in the system, it absorbs any increase in the total replicator concentration, while the concentrations of the sub-exponential replicators remain unchanged. In the context of prebiotic evolution, this means that in high-concentration local environments, an exponential replicator can reach disproportionately high concentrations even if its copying rate is lower than that of the sub-exponential replicators. In a variable environment, this can lead to the eventual stochastic extinction of its competitors, with the exponentially growing species taking over the community.

## Author summary

Synthetic chemical replicators often grow sub-exponentially due to self-inhibition induced by the reversible association between a template and its copy. This contrasts with irreversible biological reproduction resulting in exponential growth. The earliest natural replicators at the dawn of evolution were likely sub-exponential, but this turns out to severely limit their capacity to evolve. Here we

**Data availability statement:** All data and computer code used in this study can be accessed from https://github.com/dysordys/parabolic.

**Funding:** The project was supported by the ERC Synergy MiniLife Project (Grant No. 101118938 to ES, AS and BK) and the National Research, Development and Innovation Office (Grant No. 152489 to AS and BK). The funders had no role in study design, data collection and analysis, decision to publish, or preparation of the manuscript.

**Competing interests:** The authors have declared that no competing interests exist.

develop a comprehensive, analytical understanding for the deterministic dynamics of a mix of competing sub-exponential and exponential replicators. We do so by combining reaction kinetics with methods from the theory of structured populations in population ecology. This allows us to see how their potential for evolution changes, thus advancing our understanding of how they could have contributed to the emergence of life.

## Introduction

The emergence of hereditary replicators was one of the key innovations that brought about the origin of life [1]. Today, replication of the genetic material is made possible by evolved enzymes, but at the dawn of evolution, these did not yet exist. Instead, there was a supposed RNA world in which chemical replicators served both as information and as function [2,3]. A useful model of such replicators, and one that can be realized in a laboratory setting, uses self-replicating oligonucleotide strands in a chemostat [4–6]. Our purpose is to understand the ecological and evolutionary dynamics of a community of such replicators to shed light on how Darwinian replicators as we know them might have emerged.

As long as individuals replicate independently of one another, their abundance will grow or shrink exponentially. In a regulated environment (be it a natural or an experimental chemostat system) offering a common resource, exponential growth leads to competitive exclusion: the replicator with the highest capacity for growth will eventually reach a growth rate of zero, with all others having negative growth and thus eventually disappearing from the system. This, of course, is the driving force behind evolution by natural selection. But inhibitory interactions between individuals may lead to departures from exponential growth. In particular, individual strands of chemical replicators can form weakly-coupled pairs, but double-stranded replicators are replicationally inert: strands can only replicate in the unpaired state. Consequently, population growth slows down with an increasing concentration of double-stranded replicators. The resulting growth, instead of being proportional to the population's concentration itself, is usually proportional to its square root, at least when concentrations are large. Growth that scales with a power less than one of the replicator concentration is often called "sub-exponential growth", with the case of square root-proportionality called "parabolic growth" in particular [4,6,7].

Even more is true if a given strand can only associate with other strands of its own kind, but not with other types of strands. In that case, the growth inhibition arising from an increasing concentration of double-stranded replicators will exclusively affect the replicator type in question. Other replicator types are unaffected, so the above mechanism becomes strict self-inhibition. Under a wide range of circumstances, this self-inhibition can prevent any one replicator type from becoming sufficiently dominant to take over the community [8,9]. As each template is now independently regulated from the others, we observe the stable coexistence [10] of many different replicator types, or molecular "species".

This raises an important theoretical problem. Taken to the extreme, the implication is that all such species may coexist with one another. This conclusion is supported by some theoretical studies as well, which have used simplified models to describe template replication dynamics [8,11]. But this implies that natural selection no longer leads to competitive exclusion: instead of the "survival of the fittest", we have the "survival of everybody" [8]. The problem of the survival of everybody recently resurfaced in the ecological literature as well. Curiously though, it did not do so as a theoretical difficulty, but as a proposed solution to the long-standing diversity-stability debate [12–17]. This debate is about whether species diversity begets community stability, or conversely: more diversity leads to a lower likelihood for stable communities. Traditional approaches [12,14–16] claim that, all other things equal, local asymptotic stability of a community can only be maintained when the number of species is not too large. But a new study [17] argues that this is no longer the case when population growth is sub-exponential (or "sublinear", in their terminology—the difference arises from whether we focus on the pattern of growth through time (sub-exponential), or the dependence of the total population growth rate on population abundance (sublinear)). In fact, sub-exponential growth reverses the relationship, making species-rich assemblages more likely to be stable. This is an actively debated [18] (and contested [19]) explanation for the stability of highly diverse systems, even though empirical evidence that populations of biological species follow sub-exponential growth is controversial [20–24]. But if one accepts this view, then sub-exponential growth seems to provide a resolution to an important ecological problem.

Two approaches are used in the literature to describe sub-exponential growth. The first is phenomenological, where the sub-exponential dependence of growth on concentration is based on empirical findings and single-equation (approximate) dynamics [4,8,11]. The second is mechanistic modeling in which sub-exponential growth is a consequence of the dynamics of the system due to self-inhibition [9,25–28]. Phenomenological models, which were originally introduced to describe template replication dynamics, run into the problem that the growth rate of any species diverges to plus infinity as its density approaches zero, which prevents any species from going extinct. This artifact results in theoretically unbounded diversity-maintaining ability and lack of evolvability.

Here we shift our focus from phenomenological to mechanistic modeling approaches, explicitly taking the possibility of association of single strands into account. While the earlier mechanistic models [9,26,27] assumed association-dissociation equilibrium or a Michaelis–Menten approximation to the concentration of double strands [25,28], we have studied the exact dynamics without such an assumption. To simplify the analysis, we assume a logistic-type regulation on the total mass of the system and fixed resource concentration in the main text (and in S1 Appendix, Section A). We show in S1 Appendix, Section B that explicitly modeling resource inflow to the system, while leading to a slightly more complicated model, is in full qualitative agreement with our simpler approach. As we demonstrate, it is nowhere near true that any assemblage of sub-exponential species will always unconditionally coexist—as already shown in the context of ligation-based replication with a fast equilibration of single- and double-stranded forms, see Ref. [28]. In some cases, such communities behave in counterintuitive ways; for instance, increasing resource availability causes a reduction in opportunities for coexistence. Most importantly, it turns out that a community comprised of only sub-exponential species is vulnerable to invasion by a regular, exponentially-growing species. Even in cases when the sub-exponential species can coexist with this exponential one, the latter has the ability to monopolize resources and thus to reach much higher concentrations than its competitors. As a result, even in the absence of direct competitive exclusion, the sub-exponential species can slowly drift towards extinction in the presence of an exponential species.

## Results

### The dynamics of template-directed replication

In formulating the dynamics of template-directed replication, we neglect the existence of complement strands—that is, we assume that replication results in an exact copy of the template. The template and the resulting copy remain associated after copying. We assume different replicator types $i = 1, 2 \ldots N$, each characterized by an (i) association rate $a_i$ between

single-stranded replicators of the same type, (ii) dissociation rate $b_i$ of double-stranded replicators, and (iii) replication rate $c_i$. Following both ecological [29,30] and chemical [9,31] terminology, we will call these types "species". Denoting a single-stranded replicator of species $i$ by $X_i$ and a double-stranded replicator by $Y_i$, the association and dissociation reactions and the replication reaction can be written as

$$2X_i \xrightarrow{a_i} Y_i,$$
$$Y_i \xrightarrow{b_i} 2X_i,$$
$$R + X_i \xrightarrow{c_i} Y_i,$$

(1)

where R stands for the common resources (monomers) used by all species $i$. Let us now denote the concentrations of single-stranded replicators, double-stranded replicators, and the resource by $x_i$, $y_i$, and $r$, respectively. We assume a mass-regulated setting where resource concentration is constant ($\mathrm{d}r/\mathrm{d}t = 0$), and there is a species-independent outflow of strands at a dynamically adjusted rate $\varphi$. This yields the dynamics of replicator species $i$:

$$\frac{\mathrm{d}x_i}{\mathrm{d}t} = -2a_i x_i^2 + 2b_i y_i - rc_i x_i - \varphi x_i,$$
$$\frac{\mathrm{d}y_i}{\mathrm{d}t} = a_i x_i^2 - b_i y_i + rc_i x_i - \varphi y_i.$$

(2)

Since it affects all species indiscriminately, the total outflow must be proportional to the overall replicator concentration $\mu = \sum_j \mu_j = \sum_j (x_j + 2y_j)$, where $\mu_j = x_j + 2y_j$ is the total concentration of species $j$. The total outflow of the system is therefore simply set to $\varphi\mu$ at any time. The total production (number of new strands per unit time) is $r\sum_j c_j x_j$. The change in the total concentration of the system is the difference between this production and the outflow of the replicators comprising the system. At equilibrium, where the system rests at a fixed point with $\mathrm{d}x_i/\mathrm{d}t = \mathrm{d}y_i/\mathrm{d}t = 0$ for all species $i$, the two are equal, and the production of each species is equal to its own outflow.

We write $\varphi$ as the current total production divided by a constant $m$:

$$\varphi = \frac{r}{m} \sum_{j=1}^{N} c_j x_j.$$

(3)

The total replicator concentration $\mu$ then follows a standard logistic equation with carrying capacity $m$ (S1 Appendix, Section A.1); thus, $\hat{\mu} = m$, where $\hat{\mu}$ is the equilibrium value of the overall concentration. In the following, we will call $\varphi$ the normalized production, and $m$ the target replicator concentration.

In the language of population ecology, Eq (2) defines the dynamics of $N$ structured populations. Based on the basic theory of structured populations [32,33], we can conclude that (i) each species eventually achieves a stable ratio of single- vs. double-stranded replicators (the stable stage distribution [32]), and (ii) at that point, its population will have a growth rate (fitness) of

$$\Lambda_i(x_i) = \frac{-(2a_i x_i + b_i + c_i r) + \sqrt{(2a_i x_i + b_i + c_i r)^2 + 4b_i c_i r}}{2}$$

(4)

(S1 Appendix, Section A.2). Full equilibrium includes the equilibrium of the ratio of single- and double-stranded replicators, so $\Lambda_i(x_i)$ provides the exact growth rate for this case. Out of equilibrium, it can be seen as a proxy for the concentration-dependence of per capita production.

As expected, $\Lambda_i(x_i)$ is a decreasing function of this concentration (S1 Appendix, Section A.2), which means that the resulting growth slows down with increasing concentration. It is therefore called sub-exponential growth. It can also be shown that for a single species at high target replicator concentrations $m$, the total production of the system is proportional to $\sqrt{m}$, in line with the experimental results [4–6] (S1 Appendix, Section A.8). That is, under such conditions we have the simple $d\mu/dt \sim \sqrt{\mu}$. The solution to this equation is quadratic in time, which is why replicators following Eq (2) are often called not just sub-exponential, but *parabolic* replicators [8,9,11,31].

There are two exceptions to the above $x_i$-dependence of Eq (4). First, $a_i$ could be zero, preventing any association between single-stranded replicators of species $i$ (but of course, the template and its copy remain coupled after replication). In that case, Eq (4) reduces to

$$\lambda_i = \frac{-(b_i + c_i r) + \sqrt{(b_i + c_i r)^2 + 4 b_i c_i r}}{2},$$

(5)

an expression independent of $x_i$, which therefore leads to long-term exponential growth. Second, at low concentrations the quadratic term $a_i x_i^2$ in Eq (2) becomes negligible, so again we recover Eq (5). This means that a sufficiently rare sub-exponential species will grow as if it were exponential, with a rate of $\lambda_i$. Ecologically, since $\lambda_i = \Lambda_i(0)$ is the largest value $\Lambda_i(x_i)$ may take, it is the intrinsic growth rate of species $i$.

The reason a positive association rate $a_i$ prevents species $i$ from growing exponentially is that the ratio $y_i/x_i$ of double- vs. single-stranded replicators eventually shifts towards replication-inert double-stranded replicators. A zero association rate, on the other hand, means that the ratio of double- vs. single-stranded replicators is independent of concentration. To distinguish between these two fundamentally different growth dynamics, we will refer to sub-exponentially growing replicators with $a_i > 0$ as S-species, and to exponentially growing ones with $a_i = 0$ as E-species.

## Ecology of sub-exponential and exponential replicators

Ecologically, a set of different S- and E-species forms a community of strands competing for shared resources. At equilibrium, the growth rate of each species must be equal to the growth rate of the system as a whole (which is the common outflow rate for all types, see S1 Appendix, Eq (A.5)), and the concentration of each species remains constant. The equilibrium single- and double-stranded replicator concentrations can be calculated analytically from the steady-state solution of Eq (2) (see S1 Appendix, Eq (A.14) and S1 Appendix, Eqs (A.24)–(A.25)). The equilibrium ratio of the concentration of single-stranded replicators and the total concentration ($\hat{x}_i/\hat{\mu}_i$) can be also obtained directly from Eq (2) by first writing $d\mu_i/dt = dx_i/dt + 2dy_i/dt$ to obtain $d\mu_i/dt = rc_i x_i - \varphi\mu_i$, and then solving this for $d\mu_i/dt = 0$:

$$\frac{\hat{x}_i}{\hat{\mu}_i} = \frac{\hat{\varphi}}{rc_i}.$$

(6)

The equilibrium ratio $\hat{y}_i/\hat{x}_i$ of double- vs. single-stranded replicators immediately follows from this, after substituting $\hat{\mu}_i = \hat{x}_i + 2\hat{y}_i$ and rearranging:

$$\frac{\hat{y}_i}{\hat{x}_i} = \frac{c_i r - \hat{\varphi}}{2\hat{\varphi}},$$

(7)

both for S- and E-species. Here, $\hat{\varphi}$ is the equilibrium value of the normalized production $\varphi$ (formulated by S1 Appendix, Eqs (A.20) and (A.22)).

If only S-species are present, $\hat{\varphi}$ is a function of the rate constants of all species (i.e., it is jointly determined by all extant species), and it is an increasing function of the resource concentration (saturating at high values of $r$) and a decreasing

function of the target replicator concentration (saturating at low values of $m$; S1 Appendix, Section A.3.1). But an E-species (indexed with $i=0$) will completely overhaul this: when present, it becomes solely responsible for determining $\hat{\varphi}$, through the simple formula $\hat{\varphi} = \lambda_0$ (S1 Appendix, Section A.3.2). That is, the equilibrium normalized production becomes equal to the intrinsic growth rate of the E-species, with the S-species playing no role whatsoever in shaping its value. From Eq (5), this outflow is still an increasing and saturating function of $r$, but unlike before, it is now independent of $m$.

Since the growth rate of an E-species is independent of its single-stranded concentration (Eq (5)), for a given resource concentration $r$, an E-species can only be at equilibrium for a single outflow rate, equal to its intrinsic growth rate. A consequence is the well-known ecological fact that two or more E-species cannot coexist: the one with the highest growth rate excludes the others [34,35]. This means that there can be at most one single resident E-species in an equilibrium community. In contrast, due to the increasing ratio of double- vs. single-stranded replicators, the growth rate of an S-species is a decreasing function of its concentration (Eq (4)). Therefore, an S-species can satisfy several equilibrium conditions imposed by the outflow $\hat{\varphi}$, by settling on different concentration levels, always matching the given equilibrium outflow. This is the mechanism behind diversity maintenance in sub-exponential systems.

As part of the investigation of the ecological robustness of the system, S1 Appendix, Section A.7 shows how small changes in the kinetic rates (caused by external factors such as temperature, $p$H, etc.) or the external parameters $m$ and $r$, affect equilibrium concentrations. In S1 Appendix, Section A.7.3, we also examine the maximum perturbation a system can tolerate without the loss of any species.

## Invading into an established community

A new species (denoted by a prime) can invade an equilibrium resident community if it can increase from low concentrations. The condition for having a positive intrinsic growth rate is

$$\lambda' > \hat{\varphi}_{\text{res}} \tag{8}$$

(S1 Appendix, Section A.4), meaning that the intrinsic growth rate $\lambda'$ of the invader must exceed the normalized production $\hat{\varphi}_{\text{res}}$ of the established community. An alternative interpretation is that the new species can invade a system at equilibrium if its intrinsic growth rate is greater than the current growth rate of the resident species (itself equal to the system's relative outflow). This is independent of whether the resident community is purely sub-exponential or contains one E-species as well. In the latter case, $\hat{\varphi}_{\text{res}} = \lambda_0$, so invasion is successful if $\lambda' > \lambda_0$. That is, the invader must grow faster than the resident E-species, with the resident S-species playing no role.

Since the intrinsic growth rate $\lambda'$ is independent of $a'$ (Eq (5)), invasion success does not depend on the association rate of the invader. Therefore the *invasiveness* of a given species is characterized solely by its dissociation and replication rates: the higher the values of $b'$ and $c'$, the higher the probability of successful invasion into an established population (Fig 1B).

A successful invasion event never decreases, and almost always increases, the equilibrium production of the system:

$$\hat{\varphi}_{\text{new}} \geq \hat{\varphi}_{\text{res}} \tag{9}$$

(S1 Appendix, Section A.5). The only exception to a strict increase is when an S-species enters a community also containing an E-species, in which case the system's production (set by the resident E-species) may remain unchanged. Fig 1A shows the relationship between $\hat{\varphi}_{\text{res}}$ and $\hat{\varphi}_{\text{new}}$ in the case of successful invasion of sub-exponential or exponential invaders when the resident community contains only S-species, or when an E-species is additionally present. Note that for a successful sub-exponential invader, the invasion criterion remains fulfilled even in the new environment ($\lambda' > \hat{\varphi}_{\text{new}}$), aligning with the results of Ref. [28] in the case of template-directed ligation with fast association-dissociation equilibration,

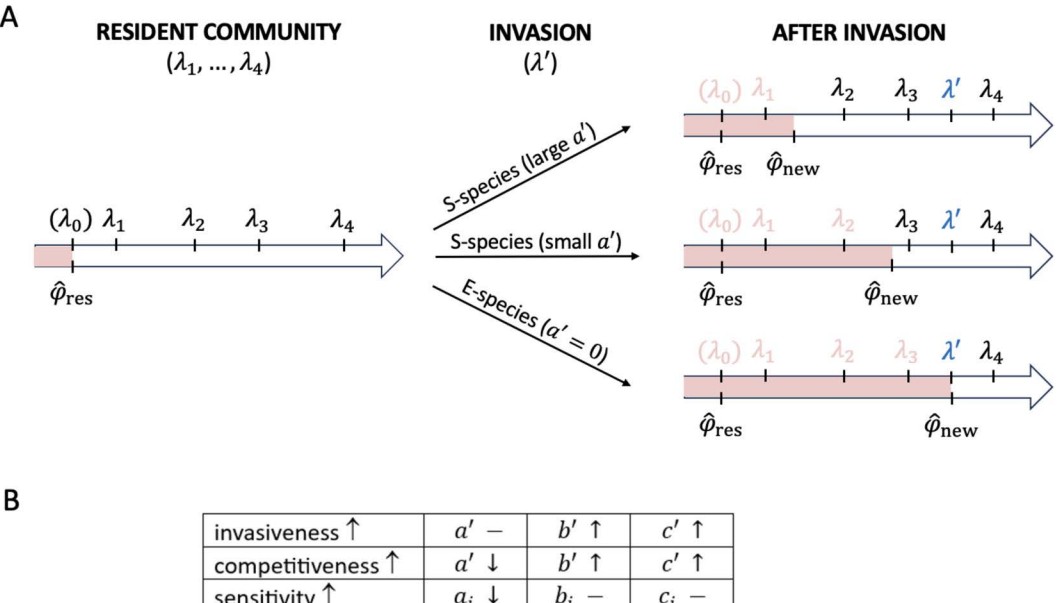

**Fig 1. General properties of invasion and exclusion as a function of the kinetic rates. (A)** Species exclusions after a successful invasion event by a weakly competitive S-species (upper arrow), a highly competitive S-species (middle arrow), or an E-species (lower arrow) into an established community. The intrinsic growth rates are $\lambda_1, \ldots, \lambda_4$ for resident S-species, $\lambda_0$ for the resident E-species (if present), and $\lambda'$ for the invader (shown in blue). Note that any E-species in the resident community must have the lowest intrinsic growth rate ($\lambda_0 < \lambda_1 < \cdots < \lambda_4$), otherwise it would have already excluded all species with even lower intrinsic growth rates. $\hat{\varphi}_{res}$ and $\hat{\varphi}_{new}$ are the normalized productions of the resident and the newly established community, respectively. E-species always fix the equilibrium normalized production at their own growth rate, so $\hat{\varphi}_{res} = \lambda_0$ if the resident population contains an E-species. If the successful invader is also an E-species, then $\hat{\varphi}_{new} = \lambda'$. $\hat{\varphi}_{new}$ is always larger than $\hat{\varphi}_{res}$; the only exception to this is when an invading S-species fails to exclude the resident E-species, yielding $\hat{\varphi}_{new} = \hat{\varphi}_{res}$. Growth rates of extirpated species, falling in the zone of exclusion given by the range between $\hat{\varphi}_{res}$ and $\hat{\varphi}_{new}$, are in red. A successfully invading new E-species always excludes at least the resident E-species, but in other cases, successful invasion need not imply extinctions. **(B)** Table shows how an invader's invasiveness (characterized by its intrinsic growth rate) and competitiveness (characterized by the width of the exclusion zone) can increase (↑), decrease (↓), or remain unaffected (-) by its kinetic rates, and how the sensitivity of a resident species $i$ (describing how fast its concentration changes with the system's production) depends on its rates.

implying that the success of invasion is equivalent to the survival in the new equilibrium. On the other hand, for a successful exponential invader $\hat{\varphi}_{new} = \lambda'$, meaning that an exponential species survives at a normalized production being equal to its intrinsic growth rate, while, according to Eq (8), it cannot invade a system with $\hat{\varphi}_{res} = \lambda'$.

Due to the increase in normalized production, a successful invasion makes all species with an intrinsic growth rate less than $\hat{\varphi}_{new}$ go extinct (S1 Appendix, Table A.1). The range between $\hat{\varphi}_{res}$ and $\hat{\varphi}_{new}$ can be interpreted as the "zone of exclusion": species whose intrinsic growth rate falls within this range are extirpated. The *competitiveness* of an invader, i.e., its ability to exclude other species, can be characterized by the width of its zone of exclusion. Since a decrease in $a'$, an increase in $b'$, and an increase in $c'$ all increase $\hat{\varphi}_{new}$ (Fig 1A, S1 Appendix, Section A.5), an invader with a lower association rate, a higher dissociation rate and/or a higher replication rate yields a wider zone of exclusion and has higher chance of excluding more resident species (Fig 1B). The result is consistent with ecological intuition: for fixed values of the parameters $b'$ and $c'$, E-species (with an association rate of zero and therefore growing fastest; Eq (4)) have the widest exclusion zones, and no S-species with the same parameter values can win against them. In general, a community is more at risk of extinctions after invasion by an E-species, whereas invasion by an S-species is more likely to lead to the retention of diversity.

Successful invasion of an E-species into a community with another E-species always leads to extinctions: at the very least, the resident E-species is replaced by the invader (Fig 1A). But if the resident community consists only of S-species,

then the invading E-species will not cause extinctions if (but only if) $\hat{\varphi}_{res} < \lambda' < \lambda_{min}$, where $\lambda_{min}$ denotes the lowest intrinsic growth rate in the resident community. In all other cases, the diversity of a sub-exponential coalition is reduced by the successful invasion of an E-species. As $\lambda'$ is an increasing function of $b'$ and $c'$, low values of both $b'$ and $c'$ of an exponential invader reduce the probability of extinctions (Fig 1B).

Successful invasion of an S-species need not lead to extinctions (Fig 1A). When invading a purely sub-exponential community, no extinctions occur if the new normalized production remains below the smallest intrinsic growth rate. When invading a community that contains an E-species, the concentration of exponential replicators is reduced, whereas the concentrations of the resident S-species remain constant due to the constant normalized production maintained by the E-species. If the sub-exponential invader does not reduce the concentration of the resident E-species to zero, then there are no extinctions, and the invading S-species increases the diversity of the system. And if the E-species goes extinct from the invasion, the new normalized production determines which other species are removed. In all these cases, high $a'$, low $b'$, and low $c'$ of an invading S-species favor diversity maintenance, leading to a lowered likelihood of extinctions (Fig 1B).

We can define the *sensitivity* of a resident species as a measure of the change in its concentration in response to either a successful invasion event or to any change in the external parameters affecting the production of the system. For a given resident species to remain at equilibrium, an increase in the equilibrium value of the normalized production $\hat{\varphi}$ must be followed by an increase in its growth rate $\Lambda_i(x_i)$. This can be achieved by decreasing its concentration $\mu_i$ because $\Lambda_i$ is a decreasing function of $x_i$. By Eq (4), the smaller the value of the association rate, the larger the concentration change required to maintain equilibrium for a given species. The most extreme case of this arises for E-species (for which $a_i = 0$), which can only equilibrate at a fixed normalized production and cannot tolerate any change in $\hat{\varphi}$ caused by the invader. For species with higher association rates, the concentration drop caused by a successful invasion or any other factors increasing the system's production is smaller. That is, the sensitivity of resident species is determined by their association rate (Fig 1B). For a more sensitive resident species (with a lower $a_i$ value), a successful invasion can more easily reduce its concentration to such small levels at which stochastic effects or other external factors already pose a high risk of extinction.

In light of the above, we can specify the largest possible subset of a given replicator library that can coexist in the long run. Starting from an arbitrary species, new species can enter until the equilibrium normalized production is smaller than the intrinsic growth rate of the invading species. Therefore, the largest long-term coexisting assembly will consist of those species that maximize $\hat{\varphi}$ (S1 Appendix, Section A.6). This set is insensitive to the order of invasions and, once formed, is evolutionarily stable (uninvadable by other species in the library). S1 Appendix, Section A.6.1 demonstrates this build-up process on a purely sub-exponential system.

## The effect of changes in the external parameters on the community composition

Here, we study the effect of changes in the target replicator concentration $m$, the resource concentration $r$, and their combined effect on the diversity-maintaining ability of the system.

**The effect of changing the target replicator concentration $m$.** As already discussed, an increase in the target replicator concentration $m$ decreases the normalized production $\hat{\varphi}$ in purely sub-exponential systems. Since successful invasion requires the intrinsic growth rate of the invader to be greater than the $\hat{\varphi}$ of the resident community, increasing $m$ increases diversity-maintaining ability for a given value of $r$. Conversely, decreasing the target replicator concentration causes gradual species loss, as shown in Fig 2 (A, top left) for a ten-species community. The target replicator concentration therefore adjusts how selective the system is. Note that a monotonic increase in diversity with increasing total replicator concentration has also been observed when modeling replication through template-directed ligation of two fragments in the Michaelis–Menten setting [28].

With only S-species, an increase in $m$ increases all equilibrium concentrations. (This is visible in S1 Appendix, Fig A.2, which shows absolute concentrations instead of the relative ones of Fig 2A.) When decreasing $m$, the species with the lowest intrinsic growth rate dies out first, since this is what the increasing normalized production of the system will first

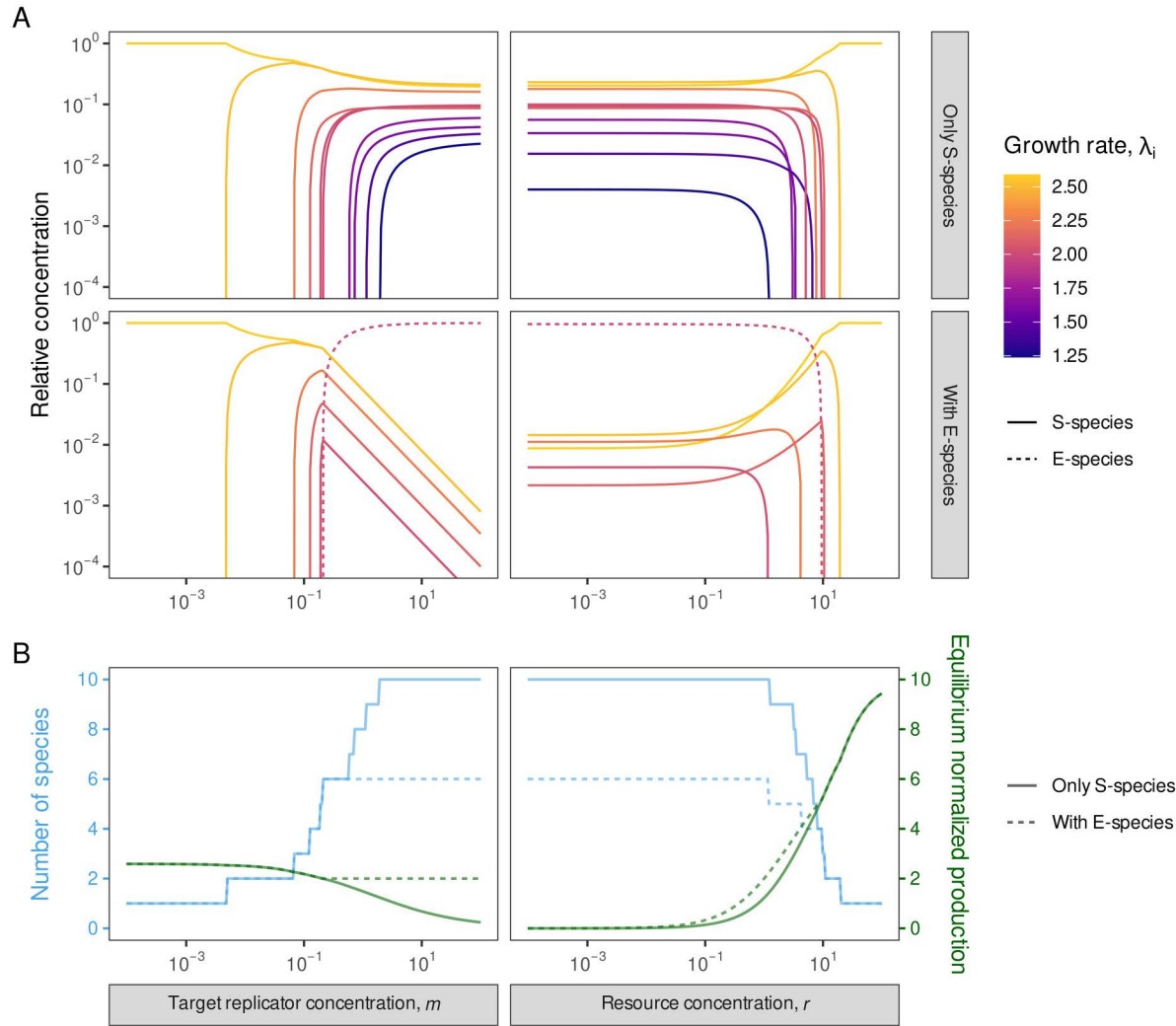

**Fig 2. Equilibria of a community of replicators, as a function of external parameters.** Panels on the left show the impact of adjusting the target replicator concentration $m$; those on the right the impact of the resource concentration $r$. **(A)** The top (bottom) panel row shows equilibrium relative concentrations with S-species only (with one E-species; dashed curve). Note the log scales of the axes. Colors indicate the intrinsic growth rates of the species, with brighter colors representing higher rates. **(B)** The number of coexisting species (blue) and the normalized production at equilibrium (green), for pure sub-exponential (solid) vs. exponential-inclusive (dashed) systems. In the left (right) column, $r=1$ ($m=2$) is fixed. In all these numerical simulations, we used a ten-species community, with $a_i$ ranging between 72.5 and 100, $b_i$ between 5.5 and 10, and $c_i$ between 1.6 and 5 for each species (see Table 1 for the detailed parameterization).

exceed. As a simple consequence, the order of extinctions follows the order of intrinsic growth rates as $m$ decreases. Additionally, the order of equilibrium concentrations at high total replicator concentrations serves as a proxy for the order of extinctions: lower concentrations tend to imply earlier extinctions. The critical target replicator concentration $m^*$ at which the first extinction occurs as $m$ decreases can also be computed (S1 Appendix, Section A.7.3).

The situation is different in the presence of an E-species. It monopolizes newly added resources, thus fixing the concentrations of extant S-species. Thus, the E-species dominates the system at high $m$ (Fig 2A, bottom left; once again, the decline in S-species concentrations is due to the relative scale). A decrease in $m$ decreases the concentration of only the E-species, so it will die out first. This is consistent with the extinction order determined by the intrinsic growth rates, as

the E-species always has the lowest intrinsic growth rate of all coexisting species (otherwise it would have excluded those with even lower intrinsic growth rates).

Fig 2 (B, left) shows a typical $\hat{\varphi}(m)$ curve (green) for a purely S-species system (solid line), and for a system of S- and an E-species (dashed line). It also shows the number of coexisting species as a function of $m$ (blue). At low target replicator concentrations, only the species with the highest intrinsic growth rate is present, with $\hat{\varphi}$ asymptotically approaching its value. The presence of an E-species prevents other species of this ten-member community from entering the system at any higher $m$ values, because the intrinsic growth rate of the non-present species is lower than that of the E-species. Nevertheless, other species (either S or E, out of the ten-member set) that have higher intrinsic growth rates than the E-species can enter the system even in the presence of the E-species. A successful invading E-species replaces the resident one, while a successful S-species need not exclude the resident E-species.

**The effect of changing the resource concentration $r$.** Just like the target replicator concentration $m$, the resource concentration $r$ also affects the diversity-maintaining ability of the system and can act as a switch between more and less selective regimes. Large values of $r$ reduce the resource limitation on growth and eventually lead to the species with the highest dissociation rate excluding all others (S1 Appendix, Section A.9), because in the abundance of resources, the replication rates ($c_i$) have no effect on the growth. For fixed $m$, a decrease in resource concentration typically increases the number of coexisting species, although not necessarily in a monotonic way. This is because in the invasion criterion both the intrinsic growth rates of Eq (5) and the normalized production (S1 Appendix, Eqs (A.20) and (A.22)) decrease towards lower resource concentrations, so a decrease in $r$ can both facilitate invasion or cause extinction.

Fig 2 (A, top right) shows the typical behavior of a purely sub-exponential system as a function of the resource concentration $r$. The behavior of the system is more complex than before: unlike in the case of $m$-dependence, varying $r$ can change the order of the growth rates and thus the order of extinctions. Furthermore, the order of the concentrations of different species at high diversity is no longer a good proxy for the order of extinctions.

When changing $r$, the presence or absence of an E-species has less effect on the dynamics than before. Both S- and E-species can enter the system at the critical resource concentration where the intrinsic growth rate exceeds the equilibrium normalized production. The E-species locks the normalized production at its intrinsic growth rate, which is an increasing function of $r$. And since the intrinsic growth rates of possible invaders are also $r$-dependent, an E-species does not lock the community composition here: unlike for $m$, further exclusions or invasions are still possible when changing $r$. Fig 2 (A, bottom right) shows the equilibrium concentrations of species as a function of resource concentration when an E-species is present in the community. In this case, after the invasion of the E-species, two other S-species can enter during the resource concentration's decrease (see also Fig 2B, right).

**Changing the target replicator and resource concentrations simultaneously.** From the above, increasing $m$ at a given $r$ increases the diversity-maintaining ability of the system by reducing the normalized production $\hat{\varphi}$, while increasing $r$ at a given $m$ typically does the opposite, though not necessarily monotonically. Fig 3 shows the typical change in the number of coexisting S-species and normalized production when varying both $m$ and $r$. When $m$ and $r$ simultaneously increase, the growth of $\hat{\varphi}$ from $r$ is somewhat offset by the increase in $m$ (Fig 3B). Meanwhile, the intrinsic growth rates are affected only by changes in $r$: when increasing both $r$ and $m$, the $\lambda$ values increase the same way as for a fixed $m$ value. Thus, while Fig 2 (B, right) shows a decay in the number of coexisting species towards larger values of $r$ at $m=2$, according to Fig 3A, a reduction of the resource limitation (an increase in $r$) combined with the growth of the system's total concentration eventually leads to a larger number of species satisfying the invasion criterion of Eq (8). It therefore yields a higher overall diversity.

## Selectivity regime shifts due to fluctuating environments

In a prebiotic context, it is natural to ask how fluctuating temperature and resource concentration affect the diversity-maintaining ability of the system. Here we investigate the replicator dynamics of Eq (2) in such a fluctuating environment. The main complication is that the equations under such continuously fluctuating conditions no longer have constant

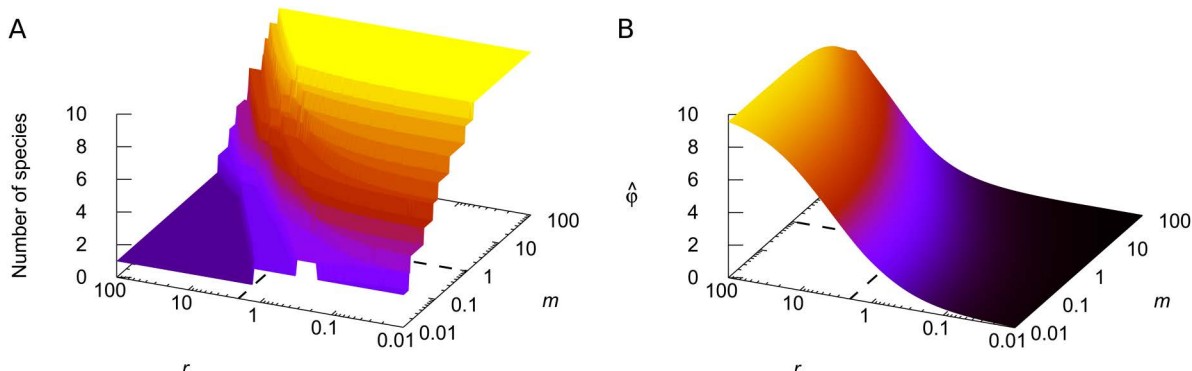

**Fig 3. The dependence of diversity and production on the external parameters. (A)** Equilibrium number of coexisting species as a joint function of the target replicator concentration $m$ and resource concentration $r$. **(B)** Normalized equilibrium production at the same settings. Dashed lines correspond to $r = 1$ and $m = 2$ used in Fig 2. We used the same 10-member community as in Fig 2.

coefficients. We simplify the problem by (i) setting the association rates of all species equal, (ii) introducing periodic fluctuations by varying this common association rate on a slow time scale to simulate temperature variation, and (iii) periodically varying $r$ on a ten times faster time scale. We introduce species, their $b$ and $c$ parameters randomly selected from the sets $b \in \{5, 5.25, \ldots, 10\}$ and $c \in \{1, 1.2, \ldots, 5\}$ (see Methods), into the system at a steady rate with low initial concentrations. We monitor the number of coexisting species in the system over time.

As seen in Fig 4, the more associative phase (scarcity of resource and/or low temperature) corresponds to a regime that allows the spread of species having suboptimal fitnesses, increasing the number of coexisting species. By contrast, the selection pressure of the less associative phase (abundance of resource and/or high temperature) reduces diversity and selects the handful of most fit species from the current community.

## Comparison of modeling frameworks

Equation (2) lies in the middle between simpler and more complicated approaches to modeling the dynamics of template-directed replication. We posit that it has the right level of complexity, still being tractable while capturing the behavior of more detailed models. For comparison, we show how our model relates to two other approaches.

First, we compare Eq (2) with the fully phenomenological model of Eigen and Schuster [31]. The model does not distinguish between single- and double-stranded replicators, and models population growth directly using an approximate form of Eq (4) that is valid for large concentrations. This results in the following equation for the concentration $z_i$ of replicator species $i$:

$$\frac{dz_i}{dt} = k_i z_i^{p_i} - \varphi z_i,$$

(10)

where $k_i$ is a rate constant, $0 < p_i < 1$ if $i$ is an S-species and $p_i = 1$ if $i$ is an E-species, $\varphi = (\sum_j k_j z_j^{p_j})/m$ is the normalized production of the whole $N$-species system, and $m$ is the target replicator concentration. The second term in Eq (10) regulates the total concentration of the system to be $m$ at equilibrium.

Equation (10) captures many of the important qualitative features of replicator ecology and evolution, such as the coexistence of many S-species and having at most one E-species in the system. However, it fails to describe the diversity and invasion patterns established above—for instance, it predicts the indiscriminate invasion success of S-species, regardless of the presence of an E-species (S1 Appendix, Section C). For these reasons, the phenomenological model given by Eq (10) is more useful as a simple illustration of the general consequences of self-inhibitory replicator

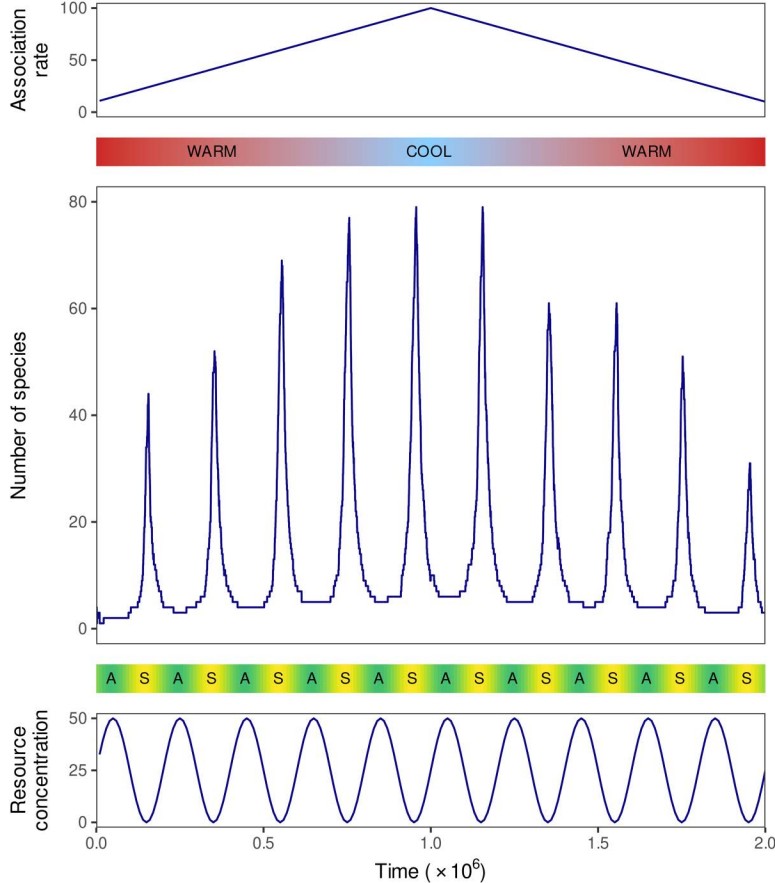

**Fig 4. The impact of the association rate and the resource concentration on the diversity-maintaining ability of a sub-exponential system.** In a variable environment, the number of coexisting S-species (middle) follows the changes of the common association rate of each species (top) and the resource concentration (bottom) in time. The number of coexisting species increases with the association rate (higher rates correspond to cooler environmental temperatures), has a local maximum when resource concentration is minimal (scarcity, S), and a local minimum when resource concentration is maximal (abundance, A). In the simulation, a new species enters the system every 100 time units with an initial concentration of $x_i = 10^{-5}$ and $y_i = 0$, having dissociation and replication rates chosen randomly from the parameter set described in Methods. A replicator species is considered established at or above a concentration of $3 \cdot 10^{-5}$; otherwise it is removed from the community when introducing the next mutant. The target replicator concentration was fixed at a constant $m = 2$ throughout the simulation.

dynamics, rather than a good quantitative description of the ecological and evolutionary patterns one expects to see in real experimental systems.

Secondly, although the assumption of mass regulation with constant resource concentration and dynamically adjusted outflow is standard in prebiotic chemistry, one could instead suppose constant outflow and time-dependent resource concentration. To see whether such resource-regulated, chemostat dynamics would qualitatively change our results, we modify Eq (2) by explicitly modeling the amount of resource $r$ flowing into the system:

$$\frac{dx_i}{dt} = -2a_i x_i^2 + 2b_i y_i - c_i r x_i - \varphi x_i,$$
$$\frac{dy_i}{dt} = a_i x_i^2 - b_i y_i + c_i r x_i - \varphi y_i,$$
$$\frac{dr}{dt} = -r \sum_{i=1}^{N} c_i x_i + (\rho - r)\varphi$$

(11)

(see Refs. [9,11,27,36] and S1 Appendix, Section B). Here $x_i$ and $y_i$ are the single- and double-stranded replicator concentrations of species $i$ as before, $\varphi$ is the chemostat's dilution rate (relative outflow volume per unit of time), and $\rho$ is the inflowing resource concentration. The last terms of the equations guarantee that the total equilibrium concentration of the system is $\rho$. The first two equations are identical to Eq (2); the final equation explicitly tracks the dynamics of the resource through time. Note that at equilibrium, the term $(\rho - r)$ equals to the equilibrium total replicator concentration $\hat{\mu}$, and thus, the equilibrium condition of the resource concentration (d$r$/d$t$=0) yields an expression analogous to Eq (3): $\varphi = \frac{\hat{r}}{\hat{\mu}} \sum_{i=1}^{N} c_i \hat{x}_i$.

According to Ref. [37], for small flow rates, the dynamics are equivalent for fixed flow systems and systems with a regulated flow that keeps concentrations constant. Based on our investigations presented in S1 Appendix, Section B, despite a third equation in Eq (11), its behavior matches that of the simpler, mass-regulated Eq (2) in general. Namely, the effect of the incoming resource concentration $\rho$ and the dilution rate $\varphi$ is analogous to that of the target replicator concentration $m$ and the fixed resource concentration $r$. Both parameters can scale the diversity-maintaining ability of the system: increasing $\rho$ ($m$) and decreasing $\varphi$ ($r$) increase the number of coexisting species (Fig 2 and S1 Appendix, Fig B.3). The strong qualitative agreement of the resource- and mass-regulated systems means that the simpler Eq (2) is a viable alternative for drawing general conclusions about coexistence and the dynamics of replicator communities.

## Discussion

In this work, we have significantly expanded the analysis of chemical replicators capable of pair formation. Taking the replicational inertness of double-stranded replicators into account, the dynamics of replicating entities changes fundamentally, making them sub-exponential, which facilitates the coexistence of different replicator species. The degree of self-inhibition, and therefore of sub-exponentiality, is governed by the association rates: high rates lead to the formation of double-stranded replicators even at low concentrations and thus more self-inhibition, whereas low association rates only lead to self-inhibition at very high concentrations, tempering the diversity-maintaining ability of the system. This confirms the result from a previous Michaelis–Menten approximation to double strand concentration [25]. In the limit of zero association, all self-inhibition is lost, and replicators follow exponential dynamics. We have established that no more than one such exponentially growing (E-)species can be present in the system at equilibrium. Further, this E-species then becomes solely responsible for determining the equilibrium point, with none of the sub-exponential (S-)species having any influence. As we have shown, an increase in the target replicator concentration $m$ increases the equilibrium concentration of the E-species only, while the sub-exponential replicators do not draw any benefit. In the context of the origin of life, this resource monopolization means that in sufficiently resource-rich local environments, an exponential replicator can enjoy a disproportionately high concentration. This is true even if the rate of synthesis $c$ of its copy is lower than that of the other replicators. Consequently, in the presence of stochastic fluctuations, the low-concentration S-species may eventually get eliminated just by chance, leaving only the E-species in the system.

We have also shown that, for fixed target replicator concentrations $m$, increasing the resource concentration $r$ tends to hinder coexistence. This may appear counterintuitive, but even more is true: the effects of increasing $r$ at a fixed $m$ are roughly the same as decreasing $m$ at a fixed $r$ (Fig 2; compare the left and right panel columns). To understand this, we start from the observation that there are two competing pathways of the formation of double-stranded replicators. The first is the pairing of two single-stranded replicators. The second is the production of a new strand, which is always bound up with its template initially. The per capita rate of the first process is $a_i x_i$; the per capita rate of the second is $c_i r$ (Eq (2)). If the concentration of type $i$ is large at a given $r$, then a large fraction of its strands is bound, association dominates replication, and sub-exponential self-inhibition is strong. But if $r$ increases for a fixed concentration of type $i$, then the number of double-stranded replicators rises due to the increased level of replication. Therefore, both high replicator concentrations and low resource concentration make self-inhibition dominant and increase diversity. That causes the right column of Fig 2 to be more or less a mirror image of the left.

As known from the ecological literature [10,34,35,38,39], the number of coexisting species is limited: at equilibrium, no more species can coexist than the number of regulating factors (regulating factors are all factors that are influenced by species density and that influence the growth rate of species [35,40]). At low replicator concentrations, self-inhibition is weak, leading to their asymptotically exponential growth. Coexistence can then only be maintained by other regulating factors. With a single regulating factor, the fastest-growing species competitively excludes all others. At high concentrations however, due to the replicational inertness of double-stranded replicators, self-inhibition can also be counted as a regulating factor, and since a given replicator can only pair up with its own copy (no cross-hybridization between different species), we automatically get one regulating factor for every S-species in the system. This opens the possibility for the coexistence of species with different intrinsic growth rates. As all species regulate their own growth, in principle, the diversity-maintaining capacity of purely parabolic systems is unlimited, meaning that any number of species can coexist. The transition between the two extremes of having at most one versus potentially infinitely many species is continuous, scaled by the association rates $a_i$. Different species can always coexist only within a given range of intrinsic growth rates (Fig 1A; see also Fig 1 in Ref. [9]), but this range expands as equilibrium species concentrations increase, allowing larger and larger differences between the intrinsic growth rates. The equilibrium species concentrations can be controlled in a natural way via external parameters (target replicator concentration $m$ and resource concentration $r$), which therefore act as control parameters regulating the diversity-maintaining ability of the system (Figs 2, 4, and S1 Appendix, Fig B.3).

These results help put in context both the validity of phenomenological approaches to the modeling of parabolic replicators [8,11,31] as well as recent ecological results on the diversity-maintaining ability of sub-exponential growth [17]. Phenomenological modeling simplifies the problem by ignoring the distinction between single and double strands, assuming that the concentration of double-stranded replicators is always large. This scenario corresponds to the high association rate limit, when self-inhibition is strong even at low concentrations. Consequently, phenomenological models of sub-exponential growth place no limits on attainable diversity as long as no E-species are present. This result is obviously an idealized limiting case; in reality, association rates are finite, and so are the zones of exclusion within which species cannot persist in the community. Unlimited coexistence is therefore not the expected outcome. The same holds when these phenomenological models are applied to ecological communities [17,19]. While the mechanisms behind sub-exponential growth in ecological systems will not match those in chemical ones, they both must share the property that growth rates cannot increase to arbitrarily high values as population densities (or concentrations) approach zero. Moving away from phenomenological models removes this artifact—and with it, the possibility for unconstrained coexistence as well.

The ability of sub-exponential growth to maintain diversity is further diminished by the possibility of cross-hybridization, a mechanism we have not considered in this work. Cross-hybridization means that two strands from different species are also capable of forming pairs [9]. The usual assumption is that the more different two strands are in their molecular construction, the weaker they are coupled (lower species-specific association and higher dissociation rates). If the molecular replicators are sufficiently long that differences of a few nucleotides are no longer detectable and such molecules pair up just as easily as strictly conspecific types, then it is only their combined concentration which grows sub-exponentially; an otherwise faster-growing strand will still outcompete slower-growing ones [9]. Note however that on average, the strength of binding between various single strands would decrease with the degree of mismatch in pairing, therefore it would still hold that sequence pairs would limit their own growth more than that of other sequences. In an ecological context, this means that simply demonstrating that species in a community individually all exhibit sub-exponential growth is insufficient. One also needs to demonstrate that this sub-exponential growth happens independently from the growth of other species, that each species is strictly *self*-inhibited. Without a demonstration that each species inhibits itself more than others, there is no guarantee that sub-exponential growth, in and of itself, promotes diversity. Developing the details of such a model is a task for the future.

Apart from ignoring cross-hybridization, our modeling approach has some other limitations. First, we have simplified the resource intake of replication by having a single resource R, whereas in reality at least two are required [4,6,41]. Each resource would pair with the template strand separately, followed by ligation. Incorporating such dynamics would likely make any analytical treatment cumbersome. That said, there is little reason to believe that this would change our results. Increasing the number of regulating factors (such as introducing more resource types or assuming four nucleobases) would increase the number of potentially coexisting E-species, but would leave the behavior of S-species qualitatively unaffected. Second, we have also assumed that the template and its copy are the same molecule. This was deliberately so in the original experiment by von Kiedrowski [4], but is not true in general. However, regarding the total concentration of a replicator species, the consideration of complement strands leads to the same dynamics (S1 Appendix, Eq (A.4)).

The problem of the coexistence of chemical/biological entities capable of replication has played a major role not only in more complex ecosystems but also since the dawn of life, during the era of prebiotic evolution. The limited possibility of coexistence is, according to our present knowledge, a common unsolved problem of the different scenarios describing the origin of life: the genome of a hypothetical minimal cell cannot be replicated in one piece at the poor replication fidelity of the prebiotic era, as genetic information is lost from the system through subsequent inaccurate replications [42,43]; while when the genome is divided into shorter, more accurately replicable parts, the difference in replication rates between the different parts causes the loss of genetic information as the slower replicating sequences are diluted out of the system. Several multilevel selection-based approaches have attempted to find a solution for the coexistence of shorter sequences replicating at different speeds (functionally coupled, surface-bound replicator communities, replicator communities encapsulated in proto-cells, etc. [44,45]), with limited success. Parabolic dynamics, which is an inherent property of template-directed replication, may also play a role here by effectively reducing competition between different species in a natural way, thereby increasing sustainable diversity and, importantly, not requiring other mechanisms (appropriate surface, proto-cell membrane formation, etc. [11]). Further studies will be needed to determine how the parabolic nature of replication dynamics can increase the amount of sustainable genetic information and help to make scenarios about the origin of life more realistic. Also, parabolic dynamics may play a role in the process of optimization in the fitness landscape. As we have demonstrated, by changing the value of certain variables that act as control parameters, the system can change its selectivity, i.e., its ability to maintain diversity, and thus switch between an exploratory phase (when the system can "spread" in the genotype space due to weak selection) and a selective phase (when the system can reach a fitness maximum, which may be far from the previous one), paving the way for fast and effective optimization.

## Methods

In our numerical investigations, we integrated the system of ODEs in Eq (2) for $10^6$ time units, which was always more than sufficient to reach equilibrium. We did this using the adaptive backward differentiation formula implemented in the `deSolve` package [46] in R [47] (all computer code to reproduce our results can be accessed from https://github.com/dysordys/parabolic). In all our simulations, we focus on a concentration regime where association is the fastest step while replication is the slowest: $a_i x_i^2 \gg b_i x_i > c_i x_i$. Namely, we let the parameters take on the following, discrete values: $a_i \in \{50, 52.5, 55, \ldots, 100\}$, $b_i \in \{5, 5.25, 5.5, \ldots, 10\}$, $c_i \in \{1, 1.2, 1.4, \ldots, 5\}$. Thus, each parameter can take on 21 different values, so together, the above possibilities lead to $21^3 = 9261$ unique permissible parameter combinations.

In purely ecological simulations where mutations are not allowed (Figs 2 and 3), we always used the same sample of ten parameter combinations to define the community's interacting species. These are listed in Table 1, together with the intrinsic growth rates $\lambda_i$ they lead to via Eq (5). In Fig 4, all the $21^2 = 441$ unique permissible $b_i - c_i$ parameter combinations participated in the simulation, while the association rate was the same for all species.

**Table 1. Kinetic rates (columns 2-4) and intrinsic growth rates (last column) of the ten replicator species used in our purely ecological numerical simulations. Species are uniquely indexed (first column). Species $i=3$ is starred because in case there is an exponential species mixed in with nine sub-exponential ones, its parameters are those of this 3rd species (but with an association rate $a_i$ of zero instead of 72.5).**

| Species index $i$ | $a_i$ | $b_i$ | $c_i$ | $\lambda_i$ |
|---|---|---|---|---|
| 1. | 72.5 | 5 | 3.4 | 1.686 |
| 2. | 72.5 | 6 | 5 | 2.262 |
| 3.* | 72.5 | 7 | 3.6 | 2.000 |
| 4. | 72.5 | 7.75 | 5 | 2.535 |
| 5. | 77.5 | 8.75 | 2 | 1.436 |
| 6. | 77.5 | 9.75 | 1.6 | 1.239 |
| 7. | 77.5 | 10 | 4.4 | 2.590 |
| 8. | 82.5 | 6 | 2.8 | 1.613 |
| 9. | 87.5 | 5.5 | 4.4 | 2.029 |
| 10. | 100.0 | 7 | 4 | 2.132 |

## Supporting information

**S1 Appendix. Detailed mathematical analysis, additional figures and tables.**
(PDF)

## Author contributions

**Conceptualization:** Géza Meszéna, Eörs Szathmáry, András Szilágyi.

**Formal analysis:** Bianka Kovács, András Szilágyi.

**Funding acquisition:** Eörs Szathmáry.

**Investigation:** Bianka Kovács, András Szilágyi.

**Methodology:** Bianka Kovács, András Szilágyi.

**Software:** György Barabás, András Szilágyi.

**Supervision:** András Szilágyi.

**Visualization:** Bianka Kovács, György Barabás, András Szilágyi.

**Writing – original draft:** Bianka Kovács, György Barabás, Géza Meszéna, Eörs Szathmáry, András Szilágyi.

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
