## [Decision Letter · Decision Letter 0]

14 Jan 2026

PCOMPBIOL-D-25-02327

The ecology and evolution of sub-exponential replicators

PLOS Computational Biology

Dear Dr. Szathmáry,

Thank you for submitting your manuscript to PLOS Computational Biology. After careful consideration, we feel that it has merit but does not fully meet PLOS Computational Biology's publication criteria as it currently stands. Therefore, we invite you to submit a revised version of the manuscript that addresses the points raised during the review process.

We look forward to receiving your revised manuscript.

Kind regards,

Jacopo Grilli

Academic Editor

PLOS Computational Biology

Zhaolei Zhang

Section Editor

PLOS Computational Biology

**Journal Requirements:**

**Reviewers' comments:**

Reviewer's Responses to Questions

**Comments to the Authors:**

Reviewer #1: The manuscript by B. Kovacs et al investigates an ecology of exponential

and sub-exponential replicators. Overall, the paper is interesting and

provide some new insights in the behavior such a system:

(a) the rules of coexistence of a (single) exponential replicator and

a set of parabolic replicators, and

(b) the effect of changing the resource concentration which, somewhat

counterintuitively, decreases diversity.

It also recovers behavior of such dynamics systems described previously in

simpler approximations of the dynamics.

Overall the manuscript is well written and the derivations in the appendix

are correct. Personally, I would have liked the main technical results to

be stated as formal Propositions to facilitate the exact comparison of the

results with the existing literature.

Regarding the behavior of the purely parabolic model, many of the results,

have been described previously in the setting of Michaelis-Menten kinetics,

i.e., assuming a fast equilibration of single-stranded and double stranded

forms. From my reading of the present manuscript, the observed behavior is

very similar. "Evolution in Systems of Ligation-Based Replicators"

(Z.Phys. 10.1524/zpch.2002.216.1.021) derives explicit survival conditions

in equilibrium, observes the monotonic increase in diversity with

increasing replicator concentration. Prop.2 in the same publication covers

the case of invasion based on an earlier result by Hofbauer at al: It implies

that the omega-limit of any trajectory starting within a subsimplex that

contains an interior fixed point is that fixed point. In other words,

Prop.2 immediately implies that invasion is successful if and only if a new

parabolic replicator is a survivor in the new equilibrium. (I suspect that

an analogous result could be proved for the author's model.)

The same publication also describes simulation results for invasion

scenarios.

As a minor note, the correspondence of a regulated flow that keeps

concentrations constant and fixed flow systems was studied "Autocatalytic

replication in a CSTR and constant organization" (Bull.Math.Biol

10.1007/s002850050155) showing that for small flow rates the dynamics

become equivalent.

In summary, I suggest to compare in more detail to the conclusions drawn in

particular from the simpler Michealis-Menten type model and to focus on on

effects provided by the present model, if any, and effects that have not be

observed already on the basis on the Michaelis-Menten approximation.

Reviewer #2: the review is uploaded as an attachment.

**Have the authors made all data and (if applicable) computational code underlying the findings in their manuscript fully available?**

Reviewer #1: Yes

Reviewer #2: Yes

PLOS authors have the option to publish the peer review history of their article (what does this mean?). If published, this will include your full peer review and any attached files.

Reviewer #1: No

Reviewer #2: **Yes:** Onofrio Mazzarisi

**Figure resubmission:**
---

## [Decision Letter · Decision Letter 1]

18 Apr 2026

Dear Professor Szathmáry,

We are pleased to inform you that your manuscript 'The ecology and evolution of sub-exponential replicators' has been provisionally accepted for publication in PLOS Computational Biology.

Best regards,

Jacopo Grilli

Academic Editor

PLOS Computational Biology

Zhaolei Zhang

Section Editor

PLOS Computational Biology

Reviewer's Responses to Questions

**Comments to the Authors:**

Reviewer #1: I have no further concerns.

Reviewer #2: The author addressed all my concerns.

**Have the authors made all data and (if applicable) computational code underlying the findings in their manuscript fully available?**

Reviewer #1: None

Reviewer #2: Yes

PLOS authors have the option to publish the peer review history of their article (what does this mean?). If published, this will include your full peer review and any attached files.

Reviewer #1: No

Reviewer #2: **Yes:** Onofrio Mazzarisi

---

## [Editor Report · Acceptance letter]

PCOMPBIOL-D-25-02327R1

The ecology and evolution of sub-exponential replicators

Dear Dr Szathmáry,

I am pleased to inform you that your manuscript has been formally accepted for publication in PLOS Computational Biology. Your manuscript is now with our production department and you will be notified of the publication date in due course.

With kind regards,

Zsofia Freund
